# A cryptic cycle in haematopoietic niches promotes initiation of malaria transmission and evasion of chemotherapy

Rebecca S. Lee[1], Andrew P. Waters [1] & James M. Brewer [1]

Blood stage human malaria parasites may exploit erythropoietic tissue niches and colonise erythroid progenitors; however, the precise influence of the erythropoietic environment on fundamental parasite biology remains unknown. Here we use quantitative approaches to enumerate *Plasmodium* infected erythropoietic precursor cells using an in vivo rodent model of *Plasmodium berghei*. We show that parasitised early reticulocytes (ER) in the major sites of haematopoiesis establish a cryptic asexual cycle. Moreover, this cycle is characterised by early preferential commitment to gametocytogenesis, which occurs in sufficient numbers to generate almost all of the initial population of circulating, mature gametocytes. In addition, we show that *P. berghei* is less sensitive to artemisinin in splenic ER than in blood, which suggests that haematopoietic tissues may enable origins of recrudescent infection and emerging resistance to antimalarials. Continuous propagation in these sites may also provide a mechanism for continuous transmission and infection in malaria endemic regions.

[1] Wellcome Centre for Molecular Parasitology, Institute of Infection, Immunity and Inflammation, College of Medical Veterinary & Life Sciences, Sir Graham Davies Building, University of Glasgow, 120 University Place, Glasgow G12 8TA Scotland, UK. Correspondence and requests for materials should be addressed to A.P.W. (email: Andy.Waters@Glasgow.ac.uk) or to J.M.B. (email: James.Brewer@Glasgow.ac.uk)

Transmission of malaria parasites (Plasmodium spp.) is effected by the small proportion of intra-erythrocytic parasites that commit to sexual development, forming either male or female gametocytes. Timely and sufficient expression of the transcription factor AP2-G is a prerequisite for commitment and is regulated epigenetically. Commitment at the level of the population appears to be a blend of stochastic activation and response to the environment[1,2]. The environmental conditions that blood stage Plasmodium are exposed to are varied. For example, many species of Plasmodium have a well-characterised proclivity to invade immature, circulating erythroid cells, suggesting that erythropoietic tissues could represent logical but under-investigated sites of colonisation[3,4]. Indeed, all stages of human infective Plasmodium falciparum have been demonstrated within the bone marrow (BM) parenchyma, with immature gametocytes having a relative enrichment within the extravascular tissue resident erythroid cells[5,6]. P. falciparum haemozoin was also detectable in BM parenchyma of an artificial humanised mouse model, suggesting that Plasmodium, or at least its products can access the BM parenchyma. Nonetheless, due to the species restriction of P. falciparum this model cannot be used to investigate the invasion of progenitor cells within the haematopoietic tissue[7]. Therefore, it remains unclear if the presence of gametocytes in intact BM parenchyma arises from preferential homing of cells containing gametocyte-committed parasites or if commitment was selectively stimulated by parasite colonisation of immature erythroid cells in vivo as has been seen in vitro[8–10].

Here we investigated in vivo the parasitisation and gametocyte formation within the erythroid precursor cells of the major sites of murine haematopoiesis (the BM and the spleen). Using a quantitative flow cytometry based approach[11], we identify the establishment of a cryptic asexual cycle within both tissues. The early reticulocytes (ER) within these tissues provide a niche for preferential gametocytogenesis, thus providing the first evidence of de novo gametocyte formation within the in vivo haematopoietic environment. Moreover, parasites within splenic ER (SER) are less sensitive to artemisinin (ART) treatment than parasites within peripheral blood. Therefore, sites of haematopoiesis may serve as origins of recrudescent infection and foci of emerging resistance to antimalarial drugs. Continuous propagation in such privileged sites may also provide a mechanism for continuing transmission and infection in settings of seasonal malarial infection.

## Results

### Plasmodium berghei[GFP] infects Ter119[+] CD44[high] erythroid cells in vivo.

To investigate the occupation of the erythropoietic niche by Plasmodium in a quantifiable manner, we adapted and optimised a flow cytometry based approach for distinguishing the various stages of the erythroid lineage[11]. The transferrin receptor (CD71) is routinely used to identify immature erythroid lineage cells[12–14]. However, we found CD71 was an inadequate marker for tissue resident immature cells, as it was also expressed on erythroid cells within the circulation (Supplementary Fig. 1A) and furthermore, was significantly downregulated on infection ($p = 0.04$, $p = 0.005$; paired $t$-test infected vs uninfected splenic cells in EryB and EryC, respectively, Supplementary Fig. 1B). We therefore developed an analytical strategy based on CD44 expression, which is expressed on erythroid differentiating cells, but downregulated during reticulocyte transmigration into the circulation[11,15] (Fig. 1a). As part of this strategy, DNA labelling was performed on CD45[low], CD11b[−], Ter119[+] BM cells to resolve nucleated erythroblasts (EB) from other erythroid cells (Fig. 1a and Supplementary Fig. 1C), where Ter119, a ubiquitous erythroid marker, was used to resolve erythroid cells from all

other lineages. This step required optimisation to allow separation of nucleated and non-nucleated erythroid cells when applied to infected samples (Supplementary Fig. 1D). Subsequently, a combination of cell size (FS-A) and CD44 expression was used to define extravascular, tissue resident ER (DNA[−]CD44[high]) from other erythroid populations in the DNA negative population, and confirmed the relative proportions previously observed (Fig. 1b)[16]. Cell sorting of the DNA[−]CD44[low] and ER directly from tissue demonstrated a distinct CD44 phenotype between reticulocytes and erythrocytes within circulation (DNA[−]CD44[low]) and erythroid cells within tissue (DNA[−]CD44[high]), circumventing the requirement to perform whole body perfusion (Supplementary Fig. 1E). Within the DNA[+] EB population, the change in cell size allowed us to define three EB subpopulations, namely, immature (largest), intermediate and mature EB (smallest), in a 1:2:4 ratio, respectively[11] (Fig. 1c). The proportions and size overlap between the ER (red) and EB populations (black) in BM and spleen were comparable to previously published data (Fig. 1a (composite) and Supplementary Fig. 1F)[17]. Furthermore, the abundance of CD44 in any of the tissue derived erythroid cellular compartments was unaffected by infection (Supplementary Table 1).

### P. berghei preferentially invades ER in haematopoietic tissues.

The refined and adapted FACS assay was used to assess the location and dynamics of non-circulating, haematopoietic tissue (spleen and BM) resident infection with constitutively green fluorescent protein (GFP) expressing P. berghei (PBGFP_CON)[18,19] (Fig. 1d). On day 7 postinfection (pi) both organs contained EB and ER erythroid cells (DNA[+]CD44[high] and DNA[−]CD44[high], respectively) that were positive by flow cytometry for GFP expressing parasites (Figs. 1d and 2a). In contrast to previous non-quantitative observations of Plasmodium spp. infections in vitro[9,10], but in accordance with histological samples[8], we did not observe a significantly higher parasitaemia within the previously defined mature EB stage compared with the other larger, more immature EB stages. Therefore, P. berghei does not appear to have a stage-specific preference for nucleated erythroid cells of different maturity in vivo (Fig. 2b). However, we did observe a significant preference for ER compared to EB in both niches and in particular (ER vs EB $p = < 0.0001$ and $p = < 0.0001$ Dunnett's multiple comparisons test; BM and spleen respectively), a significant preference for SER compared with all other non-circulating erythroid compartments ($p = < 0.0001$ One-way ANOVA) (Fig. 2a). This was reflected in the relative selectivity of P. berghei for BM EB: BM ER: spleen EB: spleen ER in the ratio 1:15:3:26, respectively (Fig. 2c). ER production results from EB division, differentiation of the most immature EB (gate 1) amplifies ER numbers by a factor of 4[15,20], therefore, an infected EB would only give rise to a single infected ER. These data therefore suggest that the preponderance of infected ER results from preferential invasion of ER rather than enhanced maturation rates of EB to ER. It has been demonstrated that Plasmodium infection causes impaired murine splenic and BM erythropoiesis, attributed to dysregulated EB amplification[21,22]. Our studies quantified this observation, identifying a significant decrease in EB numbers within the BM on day 7 pi compared with naive mice, with a similar trend observed in the spleen (Fig. 2d). Despite P. berghei infection resulting in reduced numbers and altered stage distribution of splenic EB (Fig. 2d, e), all EB stages remained susceptible to infection (Fig. 2f), although SER continued to contribute the largest extravascular parasite burden (Fig. 2g) consistent with the larger splenic erythropoietic mass. Similar infection characteristics were observed in outbred NIH mice indicating these effects were independent of genetic background (Supplementary Fig. 2A−H). Finally, mechanical passage

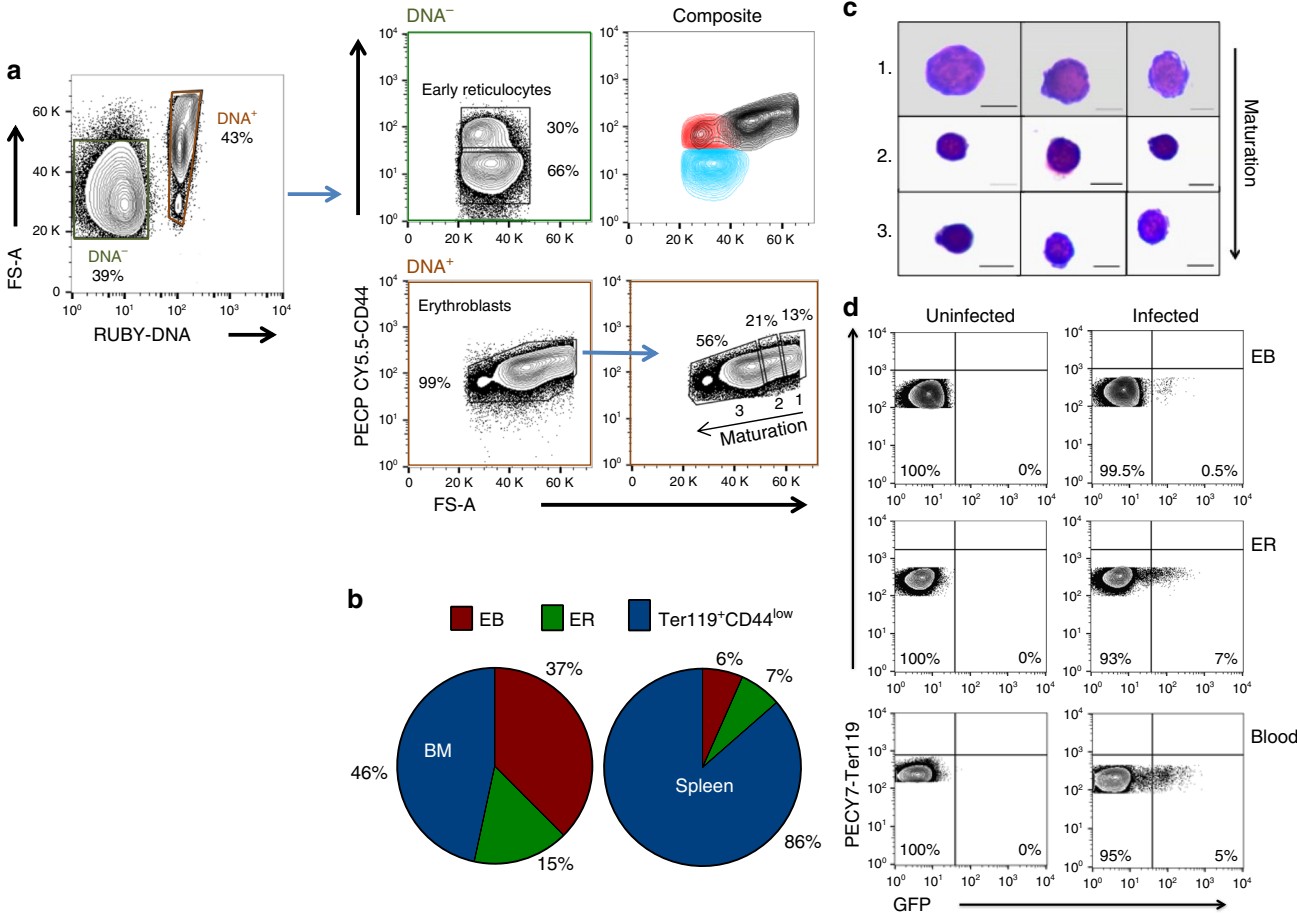

**Fig. 1** Flow analysis robustly detects *P. berghei*[GFP]-infected, Ter119[+] CD44[high] erythroid cells. **a** Representative FACS gating strategy for bone marrow (BM) and splenic resident erythroid precursors in uninfected and *P. berghei*-infected animals. DNA[−] and DNA[+] erythroid cells were gated against CD44 and FS-A to resolve CD44[high] early reticulocytes (ER) cells and erythroblasts (EB), respectively. The DNA[−]Ter119[+]CD44[low] gate was defined by the CD44 expression of erythroid cells within peripheral blood. The EB were gated in a 1:2:4 ratios to resolve immature, intermediate and mature stages, labelled 1, 2 and 3, respectively. Overlaying the DNA[−] and DNA[+] populations (composite; top right hand side of quadrant) reveals an overlap in early reticulocyte and erythroblast size. **b** To confirm erythroblast gates contained purely nucleated cells, populations were sorted from gates 1–3, stained with May-Grunwald and examined by light microscopy. Three representative cells from each group from different sorts are shown. Scale bar: 6 μm. **c** Average percentage contribution of EB, ER and Ter119[+]CD44[low] cells to the total Ter119[+] cell population in the BM and spleen of BALB/c mice. **d** Identification of GFP expressing *P. berghei* parasites (PBGFP[con]) in populations of EB, ER and peripheral blood erythrocytes (P) gated according to the developed protocol

of FACS-purified GFP[+] ER and EB to naive mice resulted in similar infection dynamics as transfer of infected blood, demonstrating the viability of the extravascular forms (Fig. 2h). Therefore, not only were ER and EB invaded by *P. berghei* but the parasite developed in these cells in vivo suggesting that a cryptic blood stage cycle of the parasite was established in these niches

***P. berghei* preferentially forms gametocytes in SER**. The observed gametocyte enrichment within BM, did not address whether parasites were sequestered from the circulation or developing de novo[6]. Interpretation may furthermore be confused by multiply invaded cells that can occur at high parasitaemia[23]. We therefore, developed tools to resolve young, actively developing gametocytes from asexual parasites for use in combination with the protocol above to characterise the parasitised erythroid lineage. The PBGFP[CON][19] parasite line was further modified to co-express red fluorescent protein (RFP) under the control of an early (8–12 hpi) gametocyte promoter, PBANKA_101270[1], generating PbGFP[CON]/RFP[GAM] parasites (Supplementary Fig.

3A−C). The half-life of RFP is typically 26 h in eukaryotic cells[24] and therefore could identify mature gametocytes. However, we have performed daily sampling to identify the first appearance of RFP positive gametocytes. When comparing the tissue resident erythroid populations, the SER was again the site of significantly higher parasitaemia as early as day 5 pi ($p = 0.0008$. One-way ANOVA)(Fig. 3a). Gametocytaemia was also greater within SER compared with BM derived ER (day 5 pi $p = <0.001$, day 6 pi $p = 0.036$, day 7 pi $p = 0.01$. One-way ANOVA), which directly translated into the highest burden of DNA[−]CD44[high], tissue resident gametocytes (day 5 pi $p = 0.014$, day 6 pi $p = 0.0029$, day 7 pi $p = 0.0026$. One-way ANOVA) (Fig. 3b, c. Supplementary Fig 3D-G). Furthermore, this ER population had significantly higher gametocytaemia (day 4 pi $p = 0.04$, day 5 pi $p = <0.001$, day 6 pi $p = 0.02$ and day 7 pi $p = 0.008$, Dunnett's multiple comparison) that preceded appearance in peripheral blood (day 4 pi) (Fig. 3b). Taken together this suggests that non-circulating ER provide a niche for the preferential formation of gametocytes that subsequently enter the circulation. We also observed asexual parasites and gametocytes within EBs from day 5 pi, as previously

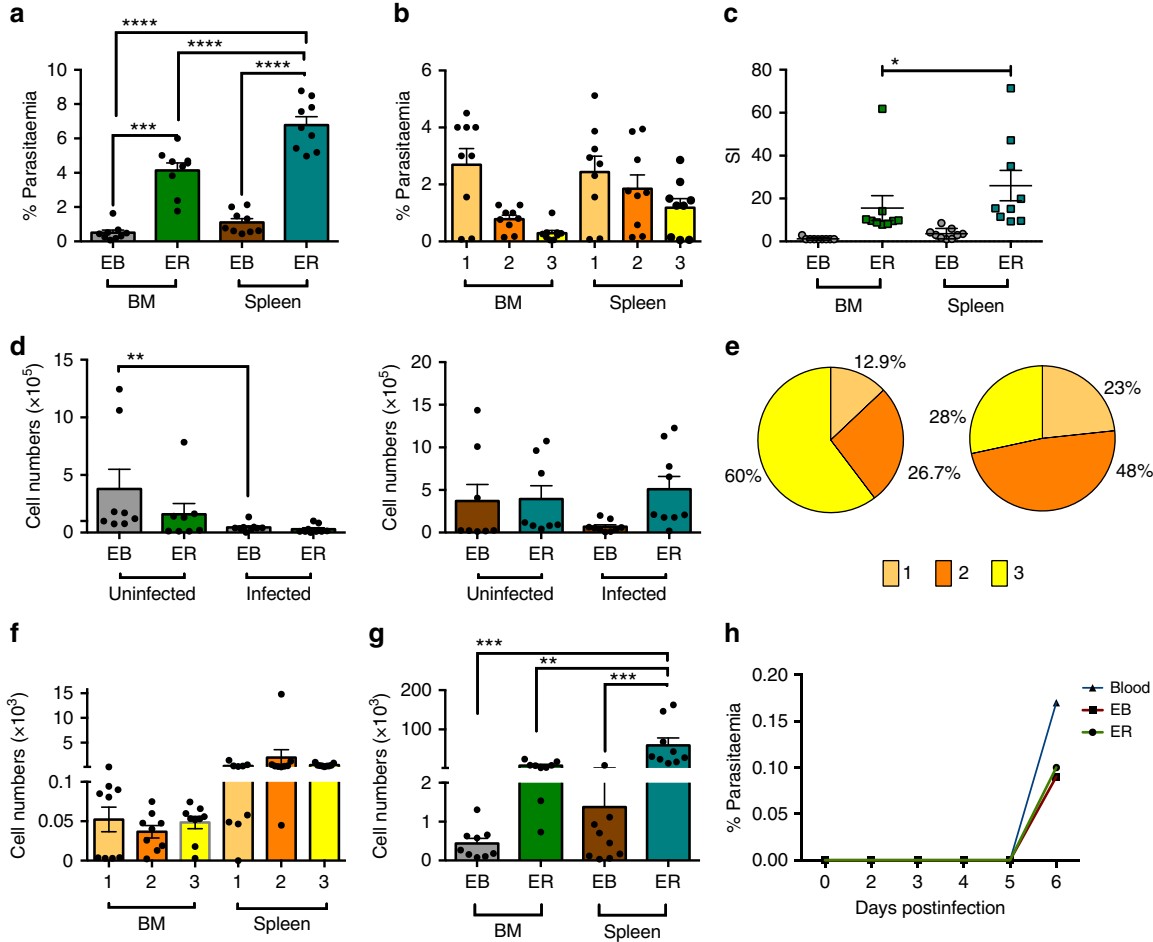

**Fig. 2** *P. berghei* preferentially invades early reticulocytes (ER) within bone marrow (BM) and particularly spleen. **a** *P. berghei* demonstrates a preference for infection of ER compared with erythroblasts (EB) in the BM and spleen of BALB/c mice at day 7 pi. Error bars ± SEM of individual values from the different experiments shown as points. **b** *P. berghei* does not show a preference for particular EB stages (1–3 as defined in Fig. 1) within BM and spleen of BALB/c mice as judged by percentage of invaded cell, under the assumption that the rate of random invasion is constant. Error bars ± SEM as in **a**. **c** The relative selectivity index for tissue resident erythroid cells was calculated for each animal by normalising the percentage parasitemia as a fraction relative to the smallest parasitemia. Error bars ± SEM as in **a**. **d** Numbers of total EB, though not ER are reduced in BM and spleen of infected BALB/c mice. Error bars ± SEM as in **a**. **e** Pie charts show the average proportion of each erythroblast stage in uninfected spleen (left) and infected spleen (right) of BALB/c mice. **f** *P. berghei* shows no preference for infection of erythroblast stages in the BM and spleen of BALB/c mice. Error bars ± SEM as in **a**. **g** *P. berghei* shows a strong preference for ER vs EB compartments in spleen and BM of BALB/c mice. Infected samples *n* = 9; uninfected *n* = 9. Error bars ± SEM as in **a**. **h** 100 GFP+ ER and EB cells were FACS sorted from spleen along with 100 GFP+ erythroid cells from the peripheral blood. *n* = 1. These were introduced into naive animals and the subsequent infections monitored by blood smears. Data from three independent experimental replications. Significant difference between groups or samples was tested using a one-way ANOVA alongside Dunnett's multiple comparisons test, except for figure **e**, which was measured using a pair *t*-test. Significance is indicated with asterisks, denoting as follows: *$p < 0.05$, **$p < 0.01$, ***$p < 0.001$, ****$p < 0.0001$

shown for *P. falciparum* in vitro (Fig. 3d)[6,9,10]. Quantitative analysis over time and assuming that only one daughter EB cell inherits any intracellular parasite present, demonstrated that the contribution of the parasitised EB population to the ensuing ER population was consistently lower than 13% in the spleen (Table 1 and Supplementary Table 2). Thus, direct invasion of ER is responsible for the majority of parasites observed in ER. Moreover, the RFP gametocyte proxy in this analysis confirms that the greatest number of gametocytes within the tissue infection originate in the ER and not the EB (Table 1). Therefore, SER is the site of both greatest non-circulating parasitaemia resulting from invasion and greatest gametocyte formation. Furthermore, quantitation of parasitised, non-circulating erythroid cells that have differentiated to produce gametocytes demonstrates that they contribute substantially to the total number of gametocytes in the infection (Supplementary Fig. 3H, I). Moreover, the dynamics of extravascular gametocyte formation demonstrates

that SER are initially, virtually the sole source of circulating gametocytes (99% on d4 of infection in NIH mice, Supplementary Fig. 3J, K). Although, this proportion declines with time (20% on day 7 pi), the additive contribution of SER remains substantial suggesting that SER are a primary source of circulating gametocytes (Supplementary Fig. 3J, K). Therefore, the erythropoietic tissue in the spleen is a significant cradle for transmission.

**ART does not clear *P. berghei* infection of SER**. The influence of a parasitised extravascular niche on effective treatment with antimalarial drugs is also obscure. Immature circulating erythrocytes (reticulocytes) are significantly more complex metabolically and enriched than their mature counterparts[25]. Consequently, antimalarials targeting parasite metabolism might become less effective due to the availability of metabolic resources that can be scavenged by the parasite from the host cell thereby

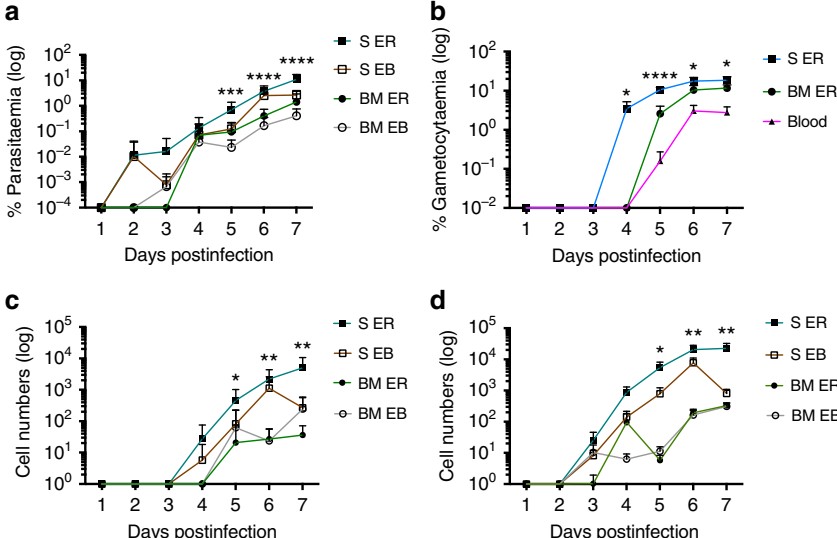

**Fig. 3** *P. berghei* gametocytes are preferentially formed within splenic early reticulocytes. Samples were taken every 24 h across an acute *P. berghei* PbGFP_{CON}/RFP_{GAM} infection and quantification of gametocytes within tissue resident erythroid cells performed by FACS. **a** The splenic ER of BALB/c mice have a higher parasitemia compared with all other tissue resident erythroid cells day 5 pi. Error bars ± SEM. **b** The gametocytemia of the splenic ER is significantly greater than peripheral blood from day 5 pi. Error bars ± SEM. The asterisks display refer to the comparison between splenic ER and peripheral blood. **c** Significantly greater numbers of gametocytes were observed in splenic ER from day 5 pi compared to all other tissue resident erythroid populations in BALB/c mice. Error bars ± SEM. **d** The abundance of asexual parasites with splenic ERs was significant within BALB/c from day 5 pi. Error bars ± SEM. Significant differences mentioned above were assessed by one-way ANOVA alongside Dunnett's multiple comparisons test and indicated with asterisks: *$p < 0.05$, **$p < 0.01$, ***$p < 0.001$, ****$p < 0.0001$

compensating for the drug-induced metabolic arrest[25]. Such metabolic or bulk masking could be potentiated in more complex erythropoietic precursors and possibly compounded through reduced drug access to the extravascular niche. As such, having established that malaria parasites form an extravascular, cryptic cycle in haematopoietic tissues, the possible influence of this niche on antimalarial efficacy was investigated measuring the survival and development of parasites in different erythroid compartments during treatment with the antimalarial, ART (Fig. 4a). Using the standard Peters' 4-day suppression test, low doses (to 10 mg/kg) of ART produced increased numbers of splenic EB/ER numbers in response to infection with *P. berghei* (Fig. 4b–e and Supplementary Fig 4A–D). At 40 mg/kg the stimulatory effect of ART on haematopoiesis was lost, mirroring previous findings on the effect of ART on primary erythropoiesis in humans[26,27]. Furthermore, the positive effect of ART on haematopoiesis could occur in the absence of infection (Supplementary Fig. 4E, F). In ART-treated mice, SER but not BM ER (Supplementary Fig. 4G, H) were colonised by both sexual and asexual *P. berghei* even at 40 mg/kg (Fig. 4f–h and Supplementary Fig. 4I) and in the absence of detectable blood stage parasitaemia (Fig. 4a). All ART-treated mice experienced recrudescence of infection (Fig. 4i) and the data suggest therefore that SER could be the source of recrudescence and furthermore, that expansion of the ER cellular niche specifically in response to ART may contribute to the failure of parasite clearance and recrudescence. Successful mechanical passage of parenchymally located parasites from the spleen of ART-treated animals demonstrated the continued viability of the parasites (Supplementary Fig. 4J).

### Discussion

Commitment to sexual development in *Plasmodium* is the result of an epigenetically regulated process that results in the production of a specific transcription factor, termed AP2-G that

rewires transcription and initiates differentiation[1,2]. However, the precise nature of the (various) triggers that result in the epigenetic shift and the pathway(s) that effect that shift are unclear but are coming into focus. Commitment is to a degree, both stochastic and environmentally sensitive, for example, positively responding to extracellular vesicles liberated from parasite-infected erythrocytes[28,29]; however, other sources of modulators and pathways of modulation of commitment may exist, for example, the serum lipid lysophosphatidylcholine, which acts as a negative regulator of commitment[30]. *P. berghei* is more likely to commit to sexual development in the extravascular niches of the BM and spleen than in circulating blood. This suggests that there is commonality shared by these two locales that exerts an influence on parasite developmental fate. We have previously demonstrated the metabolomes of maturing reticulocytes and mature erythrocytes are quite distinct and that the former is both metabolically more complex and enriched[25]. *P. berghei* preferentially invades reticulocytes and has a consistently high rate of commitment to gametocytogenesis[31] that might be influenced by the host cell metabolome. Commitment to gametocytogenesis might be enhanced in extravascular locales either through the direct influence of the erythroid precursors (metabolome) or through as yet unidentified features of these niches. Importantly, the proportion of gametocytes produced in the haematopoietic tissues is significant providing 20% of the number of bloodstream gametocytes on day 7 of infection. Indeed, the dynamics and extent of extravascular gametocyte commitment is consistent with their being the greatest source of (circulating and transmissible) gametocytes in the early phase (day 4) of infection. The application of cell fate marking technologies will allow an accurate estimation of the contribution of parasites of extravascular origin to transmission. These data underscore the continuing utility of rodent models of malaria. It is not possible to assay human haematopoietic tissue as performed here and

**Table 1 Mean percentage contribution of EB parasites to the ER asexual parasite population 24 h later in BALB/c mice**

| Days postinfection | Contribution of EB to ER asexual parasite (%) ± SD | Contribution of EB to ER gametocytes (%) ± SD |
|---|---|---|
| 2 | ND | ND |
| 3 | ND | ND |
| 4 | 0.34 ± 0.6 | ND |
| 5 | 2.56 ± 3.89 | 0.8 ± 2.3 |
| 6 | 2.28 ± 1.24 | 1.6 ± 2.6 |
| 7 | 12.83 ± 9.88 | 25 ± 24 |

All time points $n = 9$; day 7 $n = 8$
Data from three independent experimental replications
ND not detected

rodent models provide proxies that allow one to indicate what may happen in the human setting. Although, the spleen is a less significant site of haematopoiesis in the human compared with the mouse, our observations in the murine spleen are generally mirrored in the murine BM, supporting their extrapolation into human BM infections. Furthermore, our findings are consistent with the previously published data showing ability of *P. falciparum* to invade human ER in vitro[1,2] and the presence of haemozoin and enrichment of gametocytes in post-mortem BM[5,8].

Failure of ART mono and combination therapy has been well documented; however, failure is preceded by a decline in parasitaemia to levels undetectable by standard diagnostic microscopy[32]. In addition, clearance by artesunate requires extension of therapy to seven consecutive days in the *P. berghei* model[33]. Cellular environment is also significant; mutant *P. berghei* deficient in components of the haemoglobin digestion pathway colonise young reticulocytes in the circulation and in that environment are partially protected from exposure to chloroquine[34]. Clearly the enhanced survival in the face of chemotherapeutic challenge afforded by colonisation of sites of extravascular haematopoiesis might impact both continued transmission recrudescence of infection in the face of drug challenge. This work defines haematopoietic parenchyma as a potential source of recrudescent parasites and emphasizes that partner drugs must be effective in this locale. In addition, this phenomenon could create an environment where drug resistant parasites might more readily evolve due to suboptimal exposure to administered drug. This phenomenon might be exacerbated by the observation that ART treatment in common with the anti-malarial pyrimethamine, can cause haemolysis[35,36], albeit through distinct pharmacological pathways[37]. Subsequent stimulation of haematopoiesis, as observed in this study, would then generate an enlarged, localised resource that would encourage continued parasite survival. Furthermore, for certain classes of drug that target intermediary metabolism or transport processes, the evolution of drug resistance in this setting might be enhanced further by the ready source of compensatory metabolites that the erythroid host cell would potentially provide[25,38]. The rodent infection model using the reporter lines of *P. berghei* and the flow-based approach for pre-erythroid lineages employed here would provide a means to directly test if new antimalarials have similar effects on haematopoietic homoeostasis and lead to reduced drug efficacy.

Extramedullary or stress haematopoiesis can occur in many tissues, particularly in the spleen during infectious disease, as a result of hereditary blood disorders, such as β-thalassaemia[39,40], or even following ART-induced haemolysis in dogs[41]. Splenomegaly is a significant and common feature of ongoing malaria infection resulting from its roles as a primary filter of parasitised erythrocytes and the seat of initial immune responses to infection, as well as parasite-induced tissue remodelling[42]. In the context of our data, such gross distortions might mask a further haematopoietic role for the human spleen with consequent active parasitism and enhanced gametocytogenesis increasing the probability of successful transmission and furthermore, survival in a seasonal transmission setting.

Taken together the findings reported here have significance for the generation of transmission forms of malaria parasites and the ability of infection to recrudesce, survive drug challenge or persist in regions of seasonal transmission with implications for human infections.

## Methods

**Mice.** All animal experiments were performed within the strict parameters defined by the Home Office and UK Animals (Scientific Procedures) Act 1986 and approved by the University of Glasgow Ethics Committee. Six to eight week old Female BALB/cOlaHsd(Harlan) and Hsd:NIH Swiss (Harlan), weighing 15–18 g were obtained from Harlan Sprague Dawley Inc.

**P. berghei parasite lines.** The characterised *P. berghei* ANKA PbGFPko230p-SMCON (507 clone1)[18] was used to infect mice and as the parental line to *P. berghei* ANKA PbGFP$_{CON}$/RFP$_{GAM}$ line.

**Seven-day infection.** Infection of laboratory animals with *P. berghei* ANKA was performed either by intra-peritoneal injection of thawed, cryopreserved stocks or intravenously (IV) using tail blood obtained from a previously infected animal within the experimental work flow. Vasodilation of uninfected mice was induced at 37 °C for 15–20 min. Mice were appropriately restrained and inoculated IV with $10^4$ parasites. Parasitaemia was monitored daily either by visual inspection of Giemsa stained thin blood smears or by flow cytometry.

**Tissue collecting.** Mice were killed by cervical dislocation (The Humane Killing of Animals under Schedule 1 to the Animals Act 1986). Death was confirmed by cardiorespiratory failure and unresponsiveness to pain stimuli. A volume of 2–5 µl of peripheral blood was collected in 1 ml of PBS (Gibco). The spleen was removed, weighed and placed in 1 ml of 1X PBS (Gibco) and transported on ice. The femur was dislocated from the pelvis and the skin and muscle removed fully from the femur and tibia before the bones were added to 1 ml PBS (Gibco) and transported on ice. The epiphysis of the femur and tibia were removed using a single edge razor blade (Swann Morton). The BM was perfused from the diaphysis and epiphysis using a 5 ml syringe (BD Plastipak) and 26 G, 0.45 × 12 mm needle (14-12117, 2019-20, Henke sass wolf) containing 3 ml 4 °C 1X PBS (Gibco). The BM and the spleen were gently dissociated into a 50 ml falcon tube (Greiner) on ice using a 40 µm cell strainer (Greiner), the plunger of a 5 ml syringe (BD Plastipak) and 4 °C PBS (Gibco). The blood, BM and spleen cells were centrifuged at 400×*g* for 10 min at 4 °C, the supernatant discarded. The BM cells were suspended in 1 ml 4 °C PBS (Gibco) and the spleen in 10 ml. A volume of 10 µl of the single cell suspension was suspended 1:1 in 0.04% trypan blue (Sigma). Viable cells were counted using a haemocytometer (Nano Entek). Approximately $3 \times 10^7$ cells of each sample were added to a 96-well plate (Costar). The plate was centrifuged at 400×*g* for 5 min and the supernatant discarded.

**Flow cytometry.** Cell pellets were suspended in 50 µl mouse Fc block (grown in house from 2.4G2 hybridoma) and incubated at 4 °C for 30 min. The cells were then stained with fluorescently labelled antibodies diluted appropriately in 50 µl mouse Fc block and Vybrant DyeCycle Ruby stain (Life Technologies) at a 1:2000 dilution (1.25 µM concentration) (Supplementary methods) before incubating in the dark at 4 °C for 30 min. The cells were washed twice at 400×*g* for 3 min with 4 °C PBS (Gibco), then stained with Fixable viability dye, eFluor 506 (ebioscience) at a 1:1000 concentration in 4 °C PBS (Gibco). The cells were incubated at 4 °C for 30 min in the dark. The cells were washed once at 400×*g* for 3 min with PBS (Gibco) and suspended in FACS buffer (0.5 M EDTA in PBS). The cell suspensions were filtered over Nitex mesh (0.2 µm mesh) into FACS tubes (Falcon). Cells were analysed on MACSquant (Miltenyi Biotec) on the low flow rate setting. Quantitative cell numbers were derived from running 10 µl of each fully stained sample on the low flow rate setting. Data were analysed with FlowJo software (Treestar inc.).

**Cell sorting.** Cells were sorted on a FACS ARIA III (BD Biosciences) using a 70 µm nozzle, into FACS tubes (Falcon) containing 1 ml 4 °C PBS (Gibco). Sulphadiazine-enriched gametocytes (Supplementary methods) were sorted using a S3 sorter (BioRad). The cells were injected IV into uninfected mice (Supplementary methods) or centrifuged at 400×*g* for 10 min and the cell pellet smeared onto a standard microscope slide. The slides were stained in Giemsa or May-Grünwald (0.25%

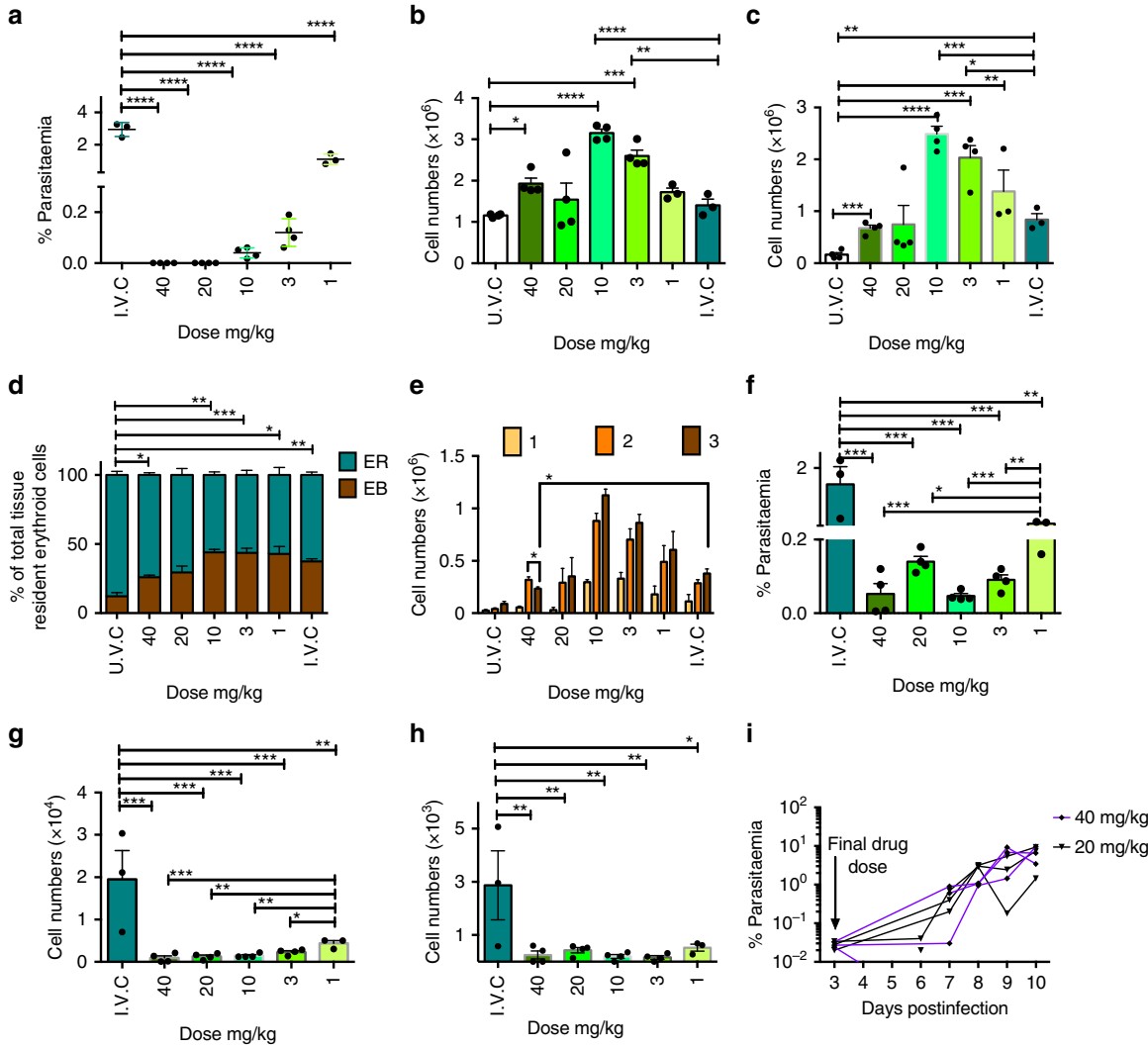

**Fig. 4** *P. berghei* infection persists within splenic early reticulocytes after ART clearance of peripheral blood infection. Groups of BALB/c mice ($n = 7$) received four doses of ART and were randomly sampled on day 4 pi, 24 h after the final doses (infected and uninfected vehicle controls; IVC and UVC, respectively. 1 mg/kg $n = 3$, 3 mg/kg $-40$mg/kg $n = 4$). **a** ART doses of 20 and 40 mg/kg cleared peripheral blood parasitemia of animals culled on day 4 pi. **b** Numbers of splenic ER in response to ART. **c** Numbers of splenic EB in response to ART. **d** The proportion of EB and ER in the spleen. **e** The proportions of EB developmental stages in response to ART. **f** Parasitemia in splenic ER during ART treatment. **g** Parasite numbers in splenic ER on day 4 pi. **h** Absolute splenic ER gametocyte numbers on day 4 pi. **i** Parasitemia of recrudescent infection observed in 20 and 40 mg/kg groups (both, $n = 3$). One experimental replication. Significant differences were assessed by one-way ANOVA alongside Dunnett's multiple comparisons test and indicated with asterisks: $*p < 0.05$, $**p < 0.01$, $***p < 0.001$

(w/v) in methanol) (Sigma-Aldrich) for 5 min, washed in 1% PBS (Gibco) being washed in water. The slides were examined under a standard light microscope using a ×100 objective and immersion oil.

**Antibodies**. PE-anti-CD44 (clone IM7) and PECY5.5-anti-CD45 (clone 30-F11) and the isotype controls were obtained from BD Biosciences and used at a final concentration of 1 ng/µl. Alexa Fluor 700 anti-CD11b (clone M1/70), eFluor 450 anti-CD71 (clone R17 217.1.14), APC-anti-CD71 (clone R17 217.1.4.), eFluor 450 anti-CD11b (clone M1/70), PECP CY5.5-anti-CD44 (clone IM7), eFluor 450 anti-CD45 (clone 30-F11), PECY7-anti-Ter119 (clone Ter119) and the isotype controls were obtained from eBioscience and used at a final concentration of 1 ng/µl.

**Selectivity index**. The selectivity index of the BM and S EB and ER populations was calculated for each individual animal as a fold increase in parasitemia. The fold increase reported for each individual was relative to the lowest percentage parasitaemia within the four tissue resident populations.

**Generation of PbGFP_{CON}/RFP_{GAM}**. DNA constructs used in the genetic modification of *P. berghei* were prepared using standard molecular biology techniques

(Supplementary methods). Transfection of PbGFP_{con} parasites with linearized constructs, positive and negative selection and cloning of the PbGFP_{CON}/RFP_{GAM} were performed as previously described[18,43,44] (Supplementary methods). Integration of the construct into the genome was checked by PCR construct (Supplementary methods & Supplementary Table 3). The dynamics of the RFP signal was investigated using a erythrocyte invasion assay was performed as described previously[45] (Supplementary methods).

**Drug dosing**. Peters' 4-day suppression test was carried out on PbGFP_{CON}/RFP_{GAM} infected or uninfected animals as previously described[46]. Stock solutions (10 and 100 mg/ml) of ART (Sigma-Aldrich Cat. No. 361593, $t_{1/2} = 23$ min ip[47]) were prepared in 100% dimethyl sulfoxide (DMSO). The ART stock solution was then diluted in sterile water with 45% cyclodextrin (Sigma-Aldrich) so the final concentration of cyclodextrin and DMSO were 10% and 3%, respectively. BALB/c mice were infected IV with $10^6$ parasites before being treated with ART (sub-cutaneous) 1 h pi and every 24 h for three additional doses. Mice in control groups received vehicle control (1X PBS or 3% DMSO and 10% cycle dextrin in sterile water). Parasitemia was monitored daily by flow cytometry. On day 4 pi para-sitaemia was measured for all mice in the group. BM and spleen were collected from 3 to 4 mice per group and flow cytometry analysis performed. The para-sitaemia of the remaining mice was monitored daily.

**Statistics**. Graphs, means, SEM and SD were calculated from three experimental repeats in triplicate unless stated otherwise using Prism 6 (GraphPad Software). Error bars are generated from SEM unless stated otherwise. Normality was tested. $p$ values were calculated using paired Student's $t$-test (comparisons of organs and erythroid compartments within a group), unpaired Student's $t$-test (comparisons between groups), one-way ANOVA in Prism 6 alongside Dunnett's multiple comparisons test (GraphPad Software). $p$ values were calculated using for non-normal data using Mann–Whitney $U$-test (GraphPad Software). Significant difference between groups or samples is indicated with asterisks, denoting as follows: $*p < 0.05$, $**p < 0.01$, $***p < 0.001$, $****p < 0.0001$.

**Data availability**. The data that support the findings of this study are available from the corresponding authors upon reasonable request.

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

## Acknowledgements

The authors would like to thank Dr. Mariana de Niz and Dr. Robyn Kent for help during the final experimental phase and Prof. Matthias Marti for extensive discussions. This work was supported by grants to A.P.W. from the Wellcome Trust (083811/Z/07/Z; 107046/Z/15/Z, WT104111AIA); to J.M.B. from Wellcome Trust (WT104111AIA). R.S.L. was supported by a MRC DTP PhD Studentship (MR/K501335/1).

## Author contributions

A.P.W and J.M.B conceived and managed the study, R.S.L performed the experiments and analyzed the data. A.P.W, J.M.B and R.S.L wrote the paper. All authors reviewed the manuscript.

## Additional information

**Competing interests:** The authors declare no competing interests.

