## [Peer Review File · Nature Communications]

Reviewers' comments:

Reviewer #1 (Remarks to the Author):

The revised submission by Lee and colleagues is much improved from the original version of the manuscript. In particular, the extended introduction and description of the experimental design makes the paper much easier for the reader to understand and interpret. The information regarding the propensity of malaria parasites to invade specific subsets of erythroid progenitors is highly relevant to our understanding of parasite replication, sexual differentiation and potentially drug resistance. This is a nice addition to the literature.

The authors have suitably addressed all the original criticisms I made to the first version of the manuscript. A few additional, minor comments are included below for the authors to consider.

1. Line 108. The authors state that *P. berghei* does not appear to have a stage specific preference for nucleated erythroid cells in vivo and cite Figure 2B. However, the bar graphs in 2B seem to show a distinct preferential trend from stage 1 to 3 in both bone marrow and spleen. Are these differences not statistically significant or should the data be interpreted some way other than simply comparing the height of the bars?

2. Lines 200-202. The authors state that mechanical passage of parenchymally located parasites from the spleen of artemisinin treated animals results in successful infection. Did the authors try passage using peripheral blood for comparison? Presumably this would either fail or require a significantly longer amount of time before patency.

3. Lines 224-226. The authors state "Importantly, the proportion of gametocytes produced in the hematopoietic tissues is significant providing 20% of the number of bloodstream gametocytes. Indeed, the dynamics and extent of extravascular gametocyte commitment is consistent with their being the greatest source of (circulating and transmissible) gametocytes." On first reading, these two sentences appear contradictory. How can "20% of bloodstream gametocytes" be the "greatest source" of gametocytes? What about the other 80%? The first sentence also seems contradictory to the abstract which states that these tissues contribute to almost the entirety of the circulating, mature gametocytes (rather than 20%). Please clarify.

Reviewer #2 (Remarks to the Author):

The author provide a revised manuscript showing that there is a cryptic malaria parasite asexual cycle in the sites of hematopoiesis (bone marrow and spleen) and that this cycle is characterized by an early commitment to gametocytogenesis. These data are particularly interesting because it is important to understand how in vitro and in vivo models of malaria differ from one another and where human parasites are likely to hide, so that better treatments can be designed. The authors have done a good job of addressing the reviewers' concerns. The strength of the manuscript is the demonstration that gametocytogenesis happens preferentially in extravascular tissues and thus may provide a reservoir of new gametocytes. Unfortunately, the weakness remains with the drug studies. While the data could be quite interesting to the community, and should be reported, the interpretation of the drug treatment studies remains difficult. This is because artemisinin, chloroquine and pyrimethamine have different serum half-lives and thus treating every 24 hour will not produce an equivalent response in the infected animal. If the drugs are to be compared, dosing needs to be adjusted so that there is equivalent exposure for different treatment groups. This may mean that the authors need to treat the artemisinin treated animals every six hours and some research may be needed to determine the appropriate dosing schedule. Alternatively, the authors could remove the comparison of artemisinin to chloroquine and pyrimethamine, and remove the speculation (lines 240 through 270) that the differences in efficacy have to do with

mechanisms of action or activity levels in extravascular niches. Likewise, the comment that erythropoietic niches provide protection against antimalarial drug activity should be removed from the abstract.

Minor points

The manuscript would be improved with the addition of subheadings with the conclusions from various sections of the results—this would make the manuscript easier to follow. For example, the authors could write at line 133 something like “gametocytogenesis is increased in early reticulocytes”.

The authors need to be very explicit about the compounds they using as well as the source (possibly even giving a catalog number) in the methods. The authors currently do not give a source for their artemisinin, and it would be imperative to know that they aren't using something like dihydroartemisinin or artesunate or artemether. They should also give the oral serum half life of the compounds somewhere in the text—there are numerous reported PK/PD studies on all of these compounds.

Reviewers' comments:

Reviewer #1 (Remarks to the Author):

The revised submission by Lee and colleagues is much improved from the original version of the manuscript. In particular, the extended introduction and description of the experimental design makes the paper much easier for the reader to understand and interpret. The information regarding the propensity of malaria parasites to invade specific subsets of erythroid progenitors is highly relevant to our understanding of parasite replication, sexual differentiation and potentially drug resistance. This is a nice addition to the literature.

The authors have suitably addressed all the original criticisms I made to the first version of the manuscript. A few additional, minor comments are included below for the authors to consider.

1. Line 108. The authors state that *P. berghei* does not appear to have a stage specific preference for nucleated erythroid cells in vivo and cite Figure 2B. However, the bar graphs in 2B seem to show a distinct preferential trend from stage 1 to 3 in both bone marrow and spleen. Are these differences not statistically significant or should the data be interpreted some way other than simply comparing the height of the bars?

These are all nucleated EB cells in Figure 2B and using a non-parametric test of multiple comparisons there is not significant difference between 1 & 2 in the BM. If the referee is looking for a comparison of ER and EB this was reported in Figure 2A. This might be a misinterpretation of our phraseology – in Fig 2B we were comparing the three stages of EB development as we defined them in Fig 1B and not establishing a preference for ER over EB. We have rephrased this section slightly to improve the clarity.

2. Lines 200-202. The authors state that mechanical passage of parenchymally located parasites from the spleen of artemisinin treated animals results in successful infection. Did the authors try passage using peripheral blood for comparison? Presumably this would either fail or require a significantly longer amount of time before patency.

The data requested by the reviewer was recorded in Figure S4R of the submitted manuscript and time to patency was indeed delayed compared with spleen. Due to the removal of other drug administration data this element of the figure is now Fig S4J.

3. Lines 224-226. The authors state “Importantly, the proportion of gametocytes produced in the hematopoietic tissues is significant providing 20% of the number of bloodstream gametocytes. Indeed, the dynamics and extent of extravascular gametocyte commitment is consistent with their being the greatest source of (circulating and transmissible) gametocytes.” On first reading, these two sentences appear contradictory. How can “20% of bloodstream gametocytes” be the “greatest

source” of gametocytes? What about the other 80%? The first sentence also seems contradictory to the abstract which states that these tissues contribute to almost the entirety of the circulating, mature gametocytes (rather than 20%). Please clarify.

We apologise for the poor writing – the statement in the abstract refers to the data presented in Figures S3K/L which demonstrate that 99% of the early wave of bloodstream gametocytes must come from the spleen when the disease is at its peak capacity for transmission. The 20% in the Discussion refers to the total number of gametocytes in the infection on d7 – the Discussion has been amended at this point in order to clarify as requested.

Reviewer #2 (Remarks to the Author):

The author provide a revised manuscript showing that there is a cryptic malaria parasite asexual cycle in the sites of hematopoiesis (bone marrow and spleen) and that this cycle is characterized by an early commitment to gametocytogenesis. These data are particularly interesting because it is important to understand how in vitro and in vivo models of malaria differ from one another and where human parasites are likely to hide, so that better treatments can be designed. The authors have done a good job of addressing the reviewers’ concerns. The strength of the manuscript is the demonstration that gametocytogenesis happens preferentially in extravascular tissues and thus may provide a reservoir of new gametocytes.

We thank the referee for this positive response to our attempts to provide an acceptable response to the first round of review.

Unfortunately, the weakness remains with the drug studies. While the data could be quite interesting to the community, and should be reported, the interpretation of the drug treatment studies remains difficult. This is because artemisinin, chloroquine and pyrimethamine have different serum half-lives and thus treating every 24 hour will not produce an equivalent response in the infected animal. If the drugs are to be compared, dosing needs to be adjusted so that there is equivalent exposure for different treatment groups. This may mean that the authors need to treat the artemisinin treated animals every six hours and some research may be needed to determine the appropriate dosing schedule. Alternatively, the authors could remove the comparison of artemisinin to chloroquine and pyrimethamine, and remove the speculation (lines 240 through 270) that the differences in efficacy have to do with mechanisms of action or activity levels in extravascular niches. Likewise, the comment that erythropoietic niches provide protection against antimalarial drug activity should be removed from the abstract.

We have taken the referee’s advice to remove the comparisons between pyrimethamine and chloroquine and artemisinin and merely report the survival of parasites in the spleen following the indicated artemisinin treatment regime for which we now provide a reference. We have amended the discussion to reflect this

and note that whilst we agree we cannot argue that this niche affords protection instead we feel can state that parasites survive in the splenic compartment in the face of artemisinin treatments that effectively (initially) clear the parasite from the blood. This has implications for sub-optimal drug treatment in the field which we note in the text and the abstract. Referee 1 appreciates this feature and the data that support it.

With regard to the superfluous discussion concerning drug treatment, we think the referee means lines (250-270 or 250-280) as 240-250 still deal with infection of the marrow. However, here we do think that the data show some points that are worth making. The passive transfer experiments and all the quantitation demonstrate *in vivo* that parasites infecting spleen haematopoietic tissue are viable and less sensitive to the action of artemisinin than parasites in the blood. Therefore we believe that once the mentions of chloroquine and pyrimethamine have been removed that the discussion remains valid. We have de-emphasised the data and conclusions in the abstract but still make mention of the phenomenon.

Minor points

The manuscript would be improved with the addition of subheadings with the conclusions from various sections of the results—this would make the manuscript easier to follow. For example, the authors could write at line 133 something like “gametocytogenesis is increased in early reticulocytes”.

4 Sub-headings have been inserted into the Results section and relate to the content of the individual figures.

The authors need to be very explicit about the compounds they using as well as the source (possibly even giving a catalog number) in the methods. The authors currently do not give a source for their artemisinin, and it would be imperative to know that they aren't using something like dihydroartemisinin or artesunate or artemether. They should also give the oral serum half-life of the compounds somewhere in the text—there are numerous reported PK/PD studies on all of these compounds.

The origins of artemisinin used in the study with catalogue number (Sigma-Aldrich #361593) and serum half-life are now given in the Methods section. As the data comparing chloroquine and pyrimethamine have been removed these compounds are not now reported.

REVIEWERS' COMMENTS:

Reviewer #1 (Remarks to the Author):

The authors have now responded to a couple rounds of review and made substantial changes to the manuscript to address various concerns. I have no additional suggestions. This is a nice addition to the literature.

Reviewer #2 (Remarks to the Author):

The authors have addressed this reviewer's concerns. I recommend acceptance.